# Predictors of Occult Metastasis and Prognostic Factors in Patients with cN0 Oral Cancer Who Underwent Elective Neck Dissection

**DOI:** 10.3390/diseases12020039

**Published:** 2024-02-12

**Authors:** Kenji Yamagata, Satoshi Fukuzawa, Atsuro Noguchi, Shohei Takaoka, Fumihiko Uchida, Naomi Ishibashi-Kanno, Hiroki Bukawa

**Affiliations:** 1Department of Oral and Maxillofacial Surgery, Institute of Medicine, University of Tsukuba, 1-1-1 Tennodai, Tsukuba 305-8575, Ibaraki, Japan; f-uchida@md.tsukuba.ac.jp (F.U.); nkanno@md.tsukuba.ac.jp (N.I.-K.); hiroki.bukawa@gmail.com (H.B.); 2Department of Oral and Maxillofacial Surgery, University of Tsukuba Hospital, 2-1-1 Amakubo, Tsukuba 305-8576, Ibaraki, Japan; sfkuzawa3104@yahoo.co.jp (S.F.); aturo.19940904@gmail.com (A.N.); sho.tmdu@gmail.com (S.T.)

**Keywords:** oral squamous cell carcinoma, elective neck dissection, supraomohyoid neck dissection, occult metastasis, neutrophil-to-lymphocyte ratio, pathological node, prognosis

## Abstract

**Simple Summary:**

Neck control is a particularly important prognostic factor, and the diagnosis of neck metastasis at the initial examination is important for oral squamous cell carcinoma (OSCC). A pathologically negative neck does not guarantee future recurrence after preventive neck dissection in patients who are clinically node-negative (cN0). We hypothesized that some factors predict poor prognosis regardless of cN0 preventive neck dissection. Among 86 patients with OSCC, occult metastases were observed in 9 (10.5%), and the prognosis of these patients was local recurrence in 6 (7.0%) and neck recurrence in 9 (10.5%). The neutrophil-to-lymphocyte ratio (NLR) and vascular invasion are good markers for detecting OM. A Cox multivariable analysis identified two independent predictors of overall survival: pathologic node and laterality of END. An independent predictive factor for disease-free survival, the surgical margin, was also identified in this study.

**Abstract:**

Elective neck dissection (END) is recommended for the management of patients with oral squamous cell carcinoma (OSCC) because of the risk of occult metastasis (OM). We hypothesized that some factors predict poor prognosis regardless of a cN0 END. This study aimed to investigate the predictors of OM and prognostic factors in patients with cN0 OSCC who underwent supraomohyoid neck dissection (SOHND). A retrospective cohort study design was created and implemented. The primary predictive variables in this study were OM and risk factors for poor prognosis after SOHND. A Cox proportional hazard model was used to adjust for the effects of potential confounders on the risk factors for poor prognoses. Among 86 patients with OSCC, OMs were observed in 9 (10.5%). The neutrophil-to-lymphocyte ratio (NLR) and vascular invasion are good markers for detecting OM. A Cox multivariable analysis identified two independent predictors of overall survival: pathologic node (pN) and laterality of END. An independent predictive factor for disease-free survival, the surgical margin, was also identified in this study. According to the pN classification, pN1 patients had a worse survival rate than pN2 patients. Therefore, in the case of pN1, regardless of being cN0, additional adjuvant therapy may be necessary.

## 1. Introduction

Supraomohyoid neck dissection (SOHND) is an elective neck dissection (END) used in some patients who have clinically node-negative (cN0) oral squamous cell carcinoma (OSCC) due to the risk of occult metastasis (OM) [1,2,3]. Although the observation policy is reasonable for appropriately selected patients who can undergo a very close follow-up [4,5], lymph node (LN) micrometastases cannot be detected using palpation, ultrasound sonography (US), computed tomography (CT), magnetic resonance imaging (MRI), or positron emission tomography-CT (PET)-CT. OMs can be detected on postoperative pathological examination using SOHND [6]. 

The incidences of metastases following neck dissection (ND) in cN0 oral squamous cell carcinoma (OSCC) are reportedly 7–33.75%. However, the early resection of LN metastases may increase the chances of survival [7,8,9,10,11,12]. Although failure after SOHND occurs in 2.1% of the cases, SOHND is appropriate for treating cN0 OSCC [8]. A systematic review and meta-analysis concluded that ENDs significantly reduce the rate of regional nodal recurrence and the death related to it in patients with OSCC [9,12]. In contrast, neck recurrence and patient survival after SOHND appear not to be related to the pathological node (pN) stage [6]; pN negativity (pN0) does not guarantee future recurrence after END in patients who are cN0 [3]. 

The investigators hypothesized that some factors predict a poor prognosis, regardless of being cN0 with END. This study specifically aimed to investigate predictors of OM and the prognostic factors in patients with cN0 OSCC who underwent SOHND and attempt to improve the prognosis of patients who are cN0. 

## 2. Materials and Methods

### 2.1. Study Design and Sample

A retrospective cohort study design was created and implemented with a population of patients undergoing SOHND for selective ND (levels I–III) for OSCC between 2011 and 2021 at the Department of Oral and Maxillofacial Surgery, University of Tsukuba Hospital (Ibaraki, Japan). All patients underwent CT and MRI with or without PET-CT and US, and the presence or absence of cervical LN metastases was determined. All patients were clinically and radiologically diagnosed as cN0. The criteria for the SOHND were applied for patients that were clinically node-negative with a performance status (PS) ≤ 1, including reconstructive surgery, tumor depth ≥ 4 mm in tongue cancer, and more than segmental resection of the mandible performed when a cervical approach was required for primary resection. All SOHNDs were performed by experienced surgeons (K.Y. and co-authors). 

Cancer was staged according to the 2017 categories of the Union for International Cancer Control categories (eighth edition), and prior cases were also restaged. The final diagnosis of the clinical stage including the N stage was decided with the tumor board consisting of an oral and maxillofacial surgeon, otorhinolaryngologist, diagnostic radiologist, and radiation oncologist. The pathological diagnosis of the primary tumor comprised histological grading, surgical margin assessment via serial sectioning, the Yamamoto–Kohama (YK) classification [13], and evaluations for lymphatic, vascular, and perineural invasions. Vascular and lymphatic invasions were evaluated using hematoxylin and eosin (HE) staining. In difficult cases, Elastica van Gieson (EVG) staining was added for vascular invasions and immunohistochemical staining with D2-40 was added for lymphatic invasions. All visible and palpable lymph nodes were extracted from ND specimens. One maximum cross section was examined for evidence of metastasis and diagnosed by experienced pathologists. Based on the pathological results, postoperative therapy was performed using chemoradiotherapy for positive margins, radiotherapy for high-risk close margins, and pN2b according to the patient’s general condition and acceptance [14,15]. 

This study was conducted in accordance with the Declaration of Helsinki and approved by the Institutional Review Board of the University of Tsukuba Hospital. The requirement for informed consent was waived due to the retrospective nature of the study (No. R05-158). 

### 2.2. Study Variables 

The primary predictive variables in this study were risk factors for poor prognosis after SOHND and OM detected on a postoperative pathological examination. The primary outcome variables were patient characteristics, including clinical tumor classification, body mass index (BMI) [16], lymphocyte count, the neutrophil-to-lymphocyte ratio (NLR) [17], and pathological results (surgical margin, positive LN, pN classification, perineural invasion, vascular invasion, lymphatic invasion, and pN stage). 

### 2.3. Data Analyses

The authors calculated the receiver operating characteristic (ROC) curves and area under the curve (AUC), sensitivity, specificity, and 95% confidence interval (CI) to determine the lymphocyte count, as well as NLR values that best defined various risk groups. Cut-off values for predicting overall survival (OS) were determined through a ROC curve analysis based on the maximum Youden index.

The patients were divided into two groups: patients with OM and those without OM. The differences between the subgroups were analyzed for significance. Subgroups were compared using the χ^2^ test and Fisher’s exact probability test. Logistic regression analysis was performed to adjust for the effects of potential confounders.

Survival curves were calculated using the Kaplan–Meier method; differences in OS rates and disease-free survival (DFS) were analyzed using the log-rank test. OS was calculated from the date of first diagnosis to death from any cause. DFS was calculated from the data of first diagnosis until no sign of cancer was found. The cut-off date for surviving patients was March 2023. Subgroup analyses with T1 and T2 early-stage patients were performed to evaluate the association between clinical factors and OS or DFS. The Cox proportional hazard model was used to adjust for the effects of potential confounders. All statistical analyses were performed using SPSS software version 29 for Macintosh (SPSS, Chicago, IL, USA). Differences with a *p*-value of less than 0.05 were considered statistically significant. 

## 3. Results

### 3.1. Patient Characteristics

In 86 patients (55 males, 31 females) with OSCC, the average age was 63.3 (29–89) years; nine patients were pN+ (10.5%). The total number of lymph node yields was 2360 with a median value of 25 (range 10–67) among all patients. The total metastatic LN count was 14; the median number of positive LNs in the nine patients was 1 (range 1–3). No extranodal extension (ENE) was detected. The pN classification was pN0 in 77 patients (89.5%), pN1 in 5 (5.8%), and pN2b in 4 (4.7%). The LN metastatic sites were level IA in one (7.1%), IB in six (42.9%), IIA in three (21.4%), and III in four (28.6%).

The primary site was the tongue in 40 patients, lower gingiva in 37, buccal mucosa in 5, and another site in 4. The T classification was T1 in 6 patients, T2 in 33, T3 in 12, T4a in 31, and T4b in 4. SOHND was performed unilaterally in 56 patients and bilaterally according to the extent of primary tumor in 7 patients. Reconstructive surgery was performed in 56 (65.1%) patients. The pathological results of histological grade; YK classification; lymph node, vascular, and perineural invasions; and surgical margins are presented in Table 1.

Adjuvant radiotherapy and chemotherapy were administered in 14 (16.3%) and 12 (14.0%) patients, respectively. The median follow-up duration was 36.4 (range 4.9–136.8) months. The prognosis of these patients was local recurrence in six patients (7.0%), neck recurrence in nine patients (10.5%), and distant metastasis in six patients (7.0%). The 12 patients (14.0%) who died from tumors included 1 due to another cause. 

### 3.2. Variables Predicting the OM for cN0 ND

#### 3.2.1. Variables for OM

The area under the ROC curve of the NLR was 0.700, with a 95% CI of 0.516–0.884 (*p* = 0.033). The cut-off value for the Youden index analysis was 1.74; the sensitivity and specificity for predicting occult LNs were 66.7% and 80.5%, respectively (Figure 1). The AUC of the lymphocyte count was 0.677 with a 95% CI of 0.515–0.839 (*p* = 0.032); the cut-off value was 1518/mm^3^ (Appendix A). For patients with OM, one (11.1%) had a lymphocyte count <1518/mm^3^, eight (88.9%), ≥1518/mm^3^; six patients (66.7%) were NLR < 1.74, and three (33.3%) were NLR ≥ 1.74. Significant differences were identified in lymphocyte count (<1518 vs. ≥1518) and NLR (<1.74 vs. ≥1.74) between those with or without OM (Table 1). Pathological vascular invasion was present in four (44.4%) patients with OM and absent in six (7.8%) patients without metastasis; a significant difference in vascular invasion was observed between the two groups (*p* < 0.001).

#### 3.2.2. Logistic Multivariate Analysis of the Parameters

A logistic multivariate analysis of the parameters selected through a univariate analysis identified two independent predictive factors for OM: NLR (≤1.74 vs. >1.74) (odds ratio [OR]: 9.674, 95% CI: 1.621–57.741; *p* = 0.013) and vascular invasion (OR: 8.548, 95% CI: 1.419–51.511; *p* = 0.019); the details are display in Table 2. These results indicate that NLR (≤1.74 vs. >1.74) and vascular invasion are good independent, associated factors of OM.

### 3.3. The Factors Affecting Prognosis after SOHND

#### 3.3.1. Association between Clinical Factors and OS and DFS (Table 3)

There was a significant difference in OS concerning T classification (*p* < 0.05) (Appendix A). In contrast, there was no significant difference in DFS concerning T classification (Appendix A). There was a significant difference in OS when patients were stratified according to pN0, 77 patients, 87.1%, and pN+, 9 for 55.6% (*p* = 0.007). Similarly, there was a significant difference in DFS when patients were stratified according to pN0, 77 patients, 85.5%, and pN+, 9 for 55.6% (*p* = 0.011). Moreover, there was a significant difference in the pN classifications between pN0, 77 patients for OS 87.1%, pN1 (5, OS 40%), and pN2b (4, OS 75.0%) (*p* = 0.006) (Figure 2) and pN0, 77 patients for DFS 85.5%, pN1 (5, OS 40.0%), and pN2b (4, OS 75.0%) (*p* = 0.009) (Appendix A). 

**Table 3 diseases-12-00039-t003:** Characteristics of patients with oral squamous cell carcinoma in relation to cumulative overall and disease-free survival.

Variables		No. of Patients (%)	OS (%)	*p* ^†^	DFS (%)	*p* ^†^
Age	≥65	44 (51.2)	80.1	0.402	82.0	0.900
Median (years)	<65	42 (48.8)	86.7		82.4	
Sex	Male	55 (64.0)	82.0	0.837	79.7	0.582
	Female	31 (36.0)	85.8		86.8	
Tobacco consumption	Present	9 (10.5)	75.0	0.891	83.3	0.683
	None	77 (89.5)	83.7		82.0	
Alcohol consumption	Present	36 (41.9)	87.2	0.516	78.8	0.559
	None	50 (58.1)	80.6		84.8	
Primary site	Tongue	40 (46.5)	81.6	0.716	80.8	0.071
	Lower gingiva	37 (43.0)	85.1		85.9	
	Buccal mucosa	5 (5.8)	100		100	
	Other	4 (1.2)	66.7		37.5	
T classification	T1	6 (7.0)	100	0.045 *	83.3	0.257
	T2	33 (38.4)	89.9		87.1	
	T3	12 (14.0)	54.9		58.3	
	T4a	31 (36.0)	84.5		85.9	
	T4b	4 (4.7)	75.0		75.0	
pN classification	N0	77 (89.5)	87.1	0.006 **	85.5	0.009 **
	N1	5 (5.8)	40.0		40.0	
	N2b	4 (4.7)	75.0		75.0	
pN	Present	9 (10.5)	55.6	0.007 **	55.6	0.011 *
	None	77 (89.5)	87.1		85.5	
Surgical margin	<5	36 (41.9)	73.7	0.043 *	69.3	0.013 *
(mm)	≥5	50 (58.1)	90.2		91.1	
Histological grade	G1	52 (60.5)	84.2	0.686	83.6	0.621
	G2	30 (34.9)	80.0		77.8	
	G3	4 (4.7)	100		100	
BMI (kg/m^2^)	<18.5	9 (10.5)	75.0	0.250	76.2	0.732
	≥18.5–<25	57 (66.3)	81.2		82.8	
	≥25	20 (23.3)	93.8		83.6	
NLR	≥1.74	65 (75.6)	84.0	0.547	82.3	0.723
	<1.74	21 (24.4)	80.4		81.0	
Laterality of neck dissection	Unilateral	79 (91.9)	86.2	0.023 *	83.1	0.339
	Bilateral	7 (8.1)	57.1		71.4	
Reconstructive surgery	Present	56 (65.1)	77.1	0.050	76.8	0.097
	None	30 (34.9)	96.2		92.8	
Lymphatic invasion	Present Absent unknown	8 (9.3) 76 (88.4) 2	62.5 87.4	0.057	62.5 85.3	0.080
Vascular invasion	Present Absent Unknown	10 (11.6) 73 (84.9) 3	68.6 88.2	0.049 *	58.3 87.9	0.006 **
Perineural invasion	Present Absent	21 (24.4) 65 (75.6)	78.6 85.6	0.648	74.1 85.4	0.420
YK classification	1 2 3 4C 4D Unknown	6 (7.0) 11 (12.8) 49 (57.0) 14 (16.3) 1 (1.2) 5	83.3 100.0 86.0 85.7 0	0.181	83.3 100.0 82.2 85.7 0	0.209

^†^ Based on log-rank test. * *p* < 0.05 Statistically significant difference, ** *p* < 0.01 Statistically significant difference. BMI, Body mass index; OS, Overall survival; DFS, Disease-free survival; T, Tumor; pN, Pathological node, YK, Yamamoto–Kohama.

There were significant differences in the OS curve when patients were divided by surgical margins <5 mm (73.7%) vs. ≥5 mm (90.2%; *p* = 0.043) or unilateral ND (86.2%) vs. bilateral ND (57.1%; *p* = 0.023) or vascular invasion present (68.6%) versus absent (88.2%; 0.049) (Appendix A). Moreover, there were significant differences in the DFS curve when patients were divided by surgical margins <5 mm (69.3%) vs. ≥5 mm (91.1%; *p* = 0.013) or vascular invasion present (58.3%) vs. absent (87.9%; 0.006) (Appendix A).

#### 3.3.2. Cox Multivariate Regression Analysis

Univariate analyses demonstrated that OS was significantly associated with pN (present vs. none), with a hazard ratio (HR) of 4.465 and 95% CI of 1.343–14.847 (*p* = 0.015). This study also identified significant associations between OS and the laterality of ND (bilateral vs. unilateral; HR: 4.104, 95% CI: 1.101–15.308; *p* = 0.036; details shown in Table 4). DFS showed a statistically significant association with pN (present vs. none), with an HR of 3.994 and 95% CI of 1.251–12.751 (*p* = 0.019). We also found significant associations between DFS and the surgical margin (<5 vs. ≥5; HR: 3.933, 95% CI: 1.232–12.554; *p* = 0.021) and vascular invasion (present vs. none; HR: 4.608, 95% CI: 1.386–15.322; *p* = 0.013; details shown in Table 5). 

The Cox multivariate analysis of the parameters selected through univariate analysis identified two independent predictive factors for OS: pN (present vs. none; HR: 4.706, 95% CI: 1.401–15.822; *p* = 0.012) and the laterality of ND (bilateral vs. unilateral; HR: 4.392, 95% CI: 1.163–16.589; *p* = 0.029). The factors for deciding the prognosis of cN0 patients who received SOHND were pN (present vs. none) and the laterality of ND (bilateral vs. unilateral). Moreover, this study identified one independent predictive factor for DFS: surgical margin (<5 vs. ≥5; HR: 3.991, 95% CI: 1.031–15.453; *p* = 0.045).

## 4. Discussion

The incidence of OM is reported to be 7–33.75% in cN0 OSCC [7,8,9,10,11,12]. This study observed OM in 9 (10.5%) of the 86 patients, consistent with previous reports. Neck control and the diagnosis of cervical LN metastasis during the initial examination are important prognostic factors [8]. LN metastasis can be detected via CT, MRI, PET-CT, and especially with US, as demonstrated for an elastic evaluation of metastatic LN [18,19,20,21]. Although a previous clinical cN0 diagnosis had been conducted, pN+ patients comprised 10.5% of the patients in our study. Therefore, the additional factors for detecting OM (NLR ≤ 1.74 and pathological vascular invasion) are suggested to be useful.

Wu et al. found that an NLR ≥ 2.95 correlated significantly with OM in tongue cancer [22]. A pretreatment NLR ≥ 2.95 correlated significantly with a larger tumor, positive LN metastasis, and perineural invasion and is an indicator of reduced survival. Our study identified no significant difference in the survival rate according to the NLR. This result is inconsistent with our results regarding the presence of OM. Their cases included tongue cancer, 21.8% OM, and more aggressive tumors compared to our cases; a possible mechanism of tumor aggressiveness is inflammation, which contributes to cancer initiation and progression and increases the NLR. In contrast, Wang et al. reported an NLR of <1.622 for distinguishing between pN0 and pN+. When OSCC did not metastasize, innate immunity was dominant, and the neutrophil count was elevated; after tumor metastasis, innate immunity weakened, and adaptive immunity became dominant. The neutrophil count decreases significantly compared to before tumor metastasis, and the mean lymphocyte count is greater than that in the no-metastasis group, resulting in a low NLR in the OM group [23]. This study identified a lymphocyte count of 1518/mm^3^ as the cut-off value in survival with ROC analysis; a significant difference in OM was identified between <1518/mm^3^ vs. ≥1518/mm^3^. Additionally, NLRs < 1.74 and ≥1.74 displayed significant differences in OM. Consistent with the results of Wang et al. [23], the lymphocyte count was higher than that in the no-metastasis group and resulted in a low NLR < 1.74 in the OM group. Therefore, an NLR of 1.74 is a good predictive marker for OM in cN0 OSCC.

Pathological vascular invasion is a risk factor for OM. A significant difference in OS was identified between the presence and absence of vascular invasion in the log-rank analysis. If vascular invasion is observed in the biopsy specimens of patients who are cN0, the possibility of OM is suggested.

Malnutrition is associated with the prognosis of malignant diseases [24,25]. The prognosis can be evaluated using the geriatric nutritional risk index (GNRI). The GNRI score (>98 vs. ≤98), along with age and stage, is a good prognostic marker in patients with OSCC. Significant differences in the OS were observed when patients were divided according to BMI (52.9 vs. 78.3% for <18.5 vs. ≥18.5 kg/m^2^; *p* = 0.002) [26]. This study identified no significant difference in the OS and DFS between BMI < 18.5 and ≥18.5 kg/m^2^. This discrepancy resulted from the background of this study group, which was cN0 and under operative general conditions.

In this study, 10.5% of the 86 patients with OSCC were pN+. Previously, pN0 was observed in 558 (69.2%) of 806 patients and pN+ in 248 (30.8%) patients. Neck recurrence occurred in 10.0% of the pN0 patients and 20.2% of the pN+ patients [10]. The discrepancy between these studies relates to the different backgrounds of the selected patients. In contrast, in our study, neck recurrence occurred in 6.5% of pN0 and 44.4% of pN+ cases, and pN+ recurrence was observed more often than that observed previously. In the systematic review and meta-analysis of neck recurrence, after pN0 disease in patients with OSCC, isolated neck recurrence was identified in 13% of pN0 neck cancer cases following END. A pN0 neck does not guarantee future recurrence. One likely source of error is the underreporting of pathological specimens. Many cases of pN0 necks contain OM that cannot be detected through bisection or HE staining alone [3]. In our study, we analyzed one maximum cross section for the detection of LN metastasis, which may have resulted in the under-detection of OMs. As a result, future studies should use serial sectioning and immunohistochemical analysis for diagnosis.

According to the pN classification, pN1 patients had a worse survival rate than pN2 patients. Radiotherapy may be administered to patients with pN2 disease (Figure 2). Haidari et al. reported that the presence of OM, regardless of being cN0, leads to decreased survival rates and may require treatment escalation and more aggressive follow-ups than cases with cN+ on preoperative diagnostic imaging [7]. In cases of pN1, the prognosis is poor regardless of being cN0; therefore, additional adjuvant therapy is suggested in the future. SOHND is effective against pN0 OSCC, relatively effective against pN1, and less effective against pN2a. The 5-year overall neck disease-free survival rates were 84.2% for pN0, 56.9% for pN1, and 27.5% for pN2a [27]. Evidence suggests that pN2b is intermediate risk, and radiotherapy is recommended; pN1 is low risk, with no need for adjuvant therapy. The European Organization for Research and Treatment of Cancer 22931 and Radiation Therapy Oncology Group 9501 studies recommend chemotherapy, radiotherapy, or their combination for high-risk (e.g., patients with positive margins or ENE) and intermediate-risk groups (e.g., patients with >2 positive LNs) [14,15]. Based on our results and previous report [7], radiotherapy for pN1 patients diagnosed with being cN0 might improve the prognosis.

The LN metastatic site was level IA in one case (7.1%), IB in six (42.9%), IIA in three (21.4%), and III in four (28.6%); no metastasis was observed in level IIB in our study. Ferreli et al. reported that the cumulative rate of OM at level IIB was 0.8%. No isolated level IIB metastases were detected among patients with a positive level IIB; all patients with nodal disease at level IIB had positive level IIA. This meta-analysis highlights how level IIB can be safely spared in SOHND patients with cN0 OSCC, reducing the risk of postoperative shoulder dysfunction [28].

The indication of END for a cN0 neck is controversial. Some investigators reported that END could prevent regional recurrence [9,12], while others considered END an aggressive treatment because of shoulder dysfunction and aesthetic side effects [29,30], and the observation policy is considered reasonable for appropriately selected patients who can undergo very close follow-ups [4,5]. In this study, we applied END to patients with reconstructive surgery, a tumor depth ≥ 4 mm in tongue cancer, and more than a segmental resection of the mandible performed when a cervical approach was required for primary resection. We found that OM occurred in 10.5% of cases, and the NLR (≤1.74 vs. >1.74) and vascular invasion are good markers for detecting OM after SOHND in patients who are cN0. Therefore, we suggest considering END for cN0 patients who meet our results; an NLR ≤ 1.74 vs. >1.74 and vascular invasion in specimen. Moreover, some clinical biomarkers to select poor prognosis patients that are cN0 are needed in a future study. 

In the Cox multivariate analysis, the prognosis of cN0 patients was determined based on the pN and the laterality of ND for OS. It was found that regional disease has an impact on OS regardless of the cN0 status. On the other hand, the surgical margin was the only an independent predictive factor for DFS. The local condition affects the results of DFS. We suggest that patients with pN+ and an advanced primary status who require bilateral ND are more likely to be affected in terms of OS compared to surgical margins. While surgical margins less than 5 mm may increase the risk of primary recurrence, additional therapy with radiotherapy and/or chemotherapy may not have an immediate effect on OS.

This study was limited by the few available cases and the bias in the patients selected for SOHND. The follow-up duration was less than the ideal 60-month number for numerous patients, and long-term follow-up data will be necessary in the future. In addition, multiple primary sites of oral cancer were included, and selecting cases with a tongue was desired. This study was retrospective; a prospective randomized multicenter study is warranted.

## 5. Conclusions

In 86 patients with cN0 OSCC, OMs were observed in 9 patients (10.5%). The prognosis was local recurrence in six patients (7.0%), neck recurrence in nine (10.5%), and distant metastasis in six (7.0%). Additionally, the NLR (≤1.74 vs. >1.74) and vascular invasion are good markers for detecting OM after SOHND in patients that are cN0. The prognostic factors for patients with cN0 OSCC were pN+ status and the laterality of ND in OS. Moreover, this study identified one independent predictive factor for DFS: the surgical margin. In the subgroup analysis of patients with early-stage T1 and T2, significant differences in OS and DFS curves were observed when patients were stratified according to the pN and laterality of ND. According to the pN classification, pN1 patients had a worse survival rate than pN2 patients. Therefore, in the case of pN1 regardless of being cN0, additional adjuvant therapy may be necessary.

## Figures and Tables

**Figure 1 diseases-12-00039-f001:**
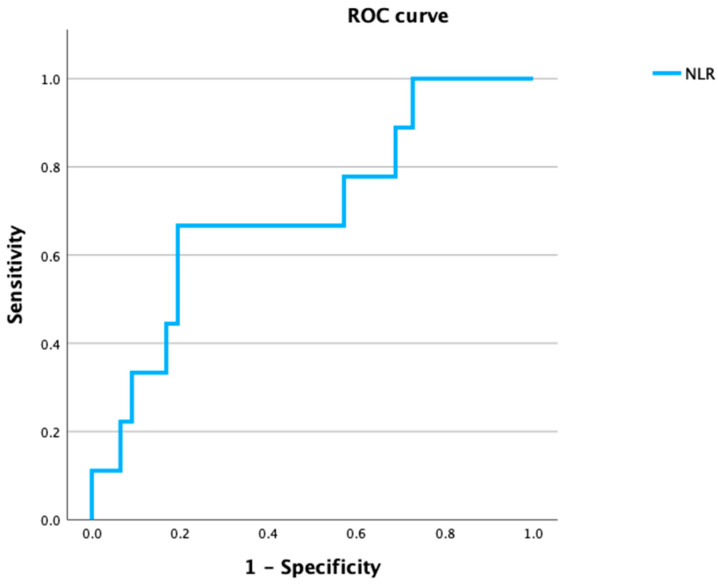
NLR with occult metastasis. The area under the receiver operating characteristic (ROC) curve was 0.700 with a 95% confidence index (CI) of 0.516–0.884 (*p* = 0.033).

**Figure 2 diseases-12-00039-f002:**
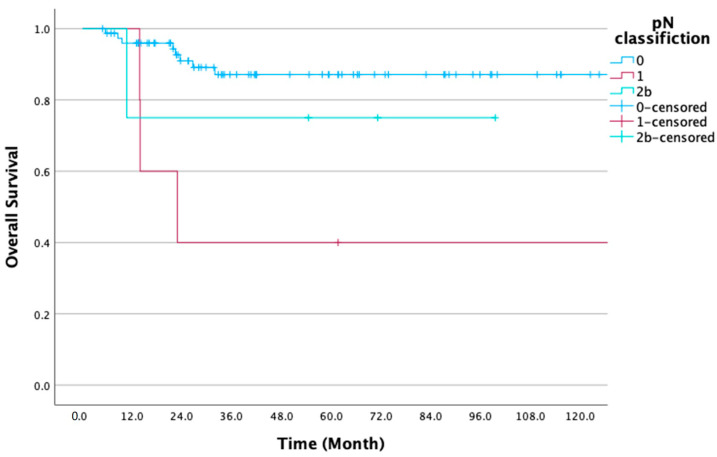
Overall survival rate according to the pN classification. There was a significant difference in the pN classification (*p* = 0.006).

**Table 1 diseases-12-00039-t001:** Clinical characteristics of the patients included in the study dichotomized with or without occult metastasis.

Variable		Total No. of Patients	No Metastasis No. of Patients (%) *n* = 77	Occult Metastasis No. of Patients (%) *n* = 9	*p*-Value
Sex	Male	55	49 (63.6)	6 (66.7)	1.000
	Female	31	28 (36.4)	3 (33.3)	
Age (years)	<65	42	38 (49.4)	4 (44.4)	1.000
	≥65	44	39 (50.6)	5 (55.6)	
BMI (kg/m^2^)	<18.5	9	8 (10.4)	1 (11.1)	0.656
	≥18.5–<25	57	50 (64.9)	7 (77.8)	
	≥25	20	19 (24.7)	1 (11.1)	
Lymphocyte count (/mm^3^)	<1518	39	38 (49.4)	1 (11.1)	0.036 *
	≥1518	47	39 (50.6)	8 (88.9)	
NLR	<1.74	21	15 (19.4)	6 (66.7)	0.006 **
	≥1.74	65	62 (80.5)	3 (33.3)	
Tabaco consumption	Present	27	26 (33.8)	1 (11.1)	0.158
	Never	59	51 (66.2)	8 (88.9)	
Alcohol consumption	Present	38	33 (42.9)	5 (55.6)	0.353
	Never	48	44 (57.1)	4 (44.4)	
Primary site	Tongue	40	36 (46.8)	4 (44.4)	0.656
	Lower gingiva	37	34 (44.2)	3 (33.3)	
	Buccal mucosa	5	4 (5.2)	1 (11.1)	
	Other	4	3 (3.9)	1 (11.1)	
T classification	T1	6	6 (7.8)	0 (0)	0.286
	T2	33	31 (40.3)	2 (22.2)	
	T3	12	9 (11.7)	3 (33.3)	
	T4a	31	28 (36.4)	3 (33.3)	
	T4b	4	3 (3.9)	1 (11.1)	
Histological grade	G1	52	47 (61.0)	5 (55.6)	0.680
	G2	30	26 (33.8)	4 (44.4)	
	G3	4	4 (5.2)	0 (0)	
YK classification	1	6	5 (6.5)	1 (11.1)	0.755
	2	11	11 (14.3)	0 (0)	
	3	49	43 (55.8)	6 (66.7)	
	4C	14	12 (15.6)	2 (22.2)	
	4D	1	1 (1.3)	0 (0)	
	Unknown	5			
Lymphatic invasion	Present	8	6 (7.8)	2 (22.2)	0.203
	Absent	76	69 (89.6)	7 (77.8)	
	Unknown	2			
Vascular invasion	Present	10	6 (7.8)	4 (44.4)	0.006 **
	Absent	73	69 (89.6)	4 (44.4)	
	Unknown	2			
Perineural invasion	Present	21	18 (23.4)	3 (33.3)	0.682
	Absent	65	59 (76.6)	6 (66.7)	

* *p* < 0.05, Statistically significant difference. ** *p* < 0.01, Statistically significant difference. BMI, Body mass index; OS, Overall survival; T, Tumor; pN, Pathological node, YK, Yamamoto–Kohama.

**Table 2 diseases-12-00039-t002:** Logistic multivariate analysis of the preoperative parameters.

	B	Wald	OR	95% CI	*p*
NLR (≤1.74 vs. >1.74)	2.269	6.199	9.674	1.621–57.741	0.013 *
Vascular invasion (Present vs. Absent)	2.146	5.482	8.548	1.419–51.511	0.019 *

* *p* < 0.05 Statistically significant difference. OR, Odds ratio; CI, Confidence interval.

**Table 4 diseases-12-00039-t004:** Univariate and multivariate Cox regression analyses for cumulative overall survival in the primary cohort.

	Univariate Analysis		Multivariate Analysis	
Variables	HR (95% CI)	*p* Values ^†^	HR (95% CI)	*p* Values ^†^
pN				
Present vs. None	4.465 (1.343–14.847)	0.015 *	4.709 (1.401–15.822)	0.012 *
Laterality of neck dissection				
Bilateral vs. Unilateral	4.104 (1.101–15.308)	0.036 *	4.392 (1.163–16.589)	0.029 *
Surgical margin (mm)				
<5 vs. ≥5	3.236 (0.973–10.767)	0.056		
T classification				
T1,2 vs. T3,4	0.362 (0.098–1.339)	0.128		
Vascular invasion				
Present vs. None	3.570 (0.922–13.822)	0.065		

**^†^** Using Cox’s proportional hazard regression. * *p* < 0.05 Statistically significant difference. BMI, Body mass index; HR, hazard ratio; CI, confidence interval.

**Table 5 diseases-12-00039-t005:** Univariate and multivariate Cox regression analyses for disease-free survival in the primary cohort.

	Univariate Analysis		Multivariate Analysis	
Variables	HR (95% CI)	*p* Values ^†^	HR (95% CI)	*p* Values ^†^
pN				
Present vs. None	3.994 (1.251–12.751)	0.019 *	1.244 (0.192–8.067)	0.819
Surgical margin (mm)				
<5 vs. ≥5	3.933 (1.232–12.554)	0.021 *	3.991 (1.031–15.453)	0.045 *
Vascular invasion				
Present vs. None	4.608 (1.386–15.322)	0.013 *	2.889 (0.526–15.882)	0.222

**^†^** Using Cox’s proportional hazard regression. * *p* < 0.05 Statistically significant difference. BMI, Body mass index; HR, hazard ratio; CI, confidence interval.

## Data Availability

Data are unavailable due to privacy or ethical restrictions.

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
