# Peer review of "Predictors of Occult Metastasis and Prognostic Factors in Patients with cN0 Oral Cancer Who Underwent Elective Neck Dissection"

_diseases, 2024, doi:10.3390/diseases12020039_

Round 1

Reviewer 1 Report

Comments and Suggestions for Authors

After reading the manuscript, I have some questions and comments for the authors to improve the manuscript.

1. The authors need to add references to support the statement of the lines 41-42 (it is also in the abstract): “Elective neck dissection (END) is recommended for the management of patients with early oral squamous cell carcinoma (OSCC) due to the risk of occult metastasis (OM).” Could the authors elaborate more, particularly regarding tumor location. Except for the lateral border of the tongue and, perhaps, floor of mouth, this statement seems not true. Moreover, define early-stage OSCC. Looking at your cohort, 54% of the patients were diagnosed with tumors at T3 and T4, which is far from early OSCC.

2. As all patients were subjected to CT and MRI +/- PET-CT and US (line 72), the term clinically NO (cN0) is inadequate. They are both clinically and radiologically N0. In this sense, 4 patients were pathologically classified as N2b (pN2b) after diagnosis of clinically and radiologically N0, meaning that they displayed lymph node metastasis (at least one ipsilateral lymph node) with at least 3 cm (>3, <6 cm). The authors should bring a constructive discussion about the inefficiency of those methods on diagnosis of positive cervical lymph nodes.

3. The recruiting period was between 2011 and 2021. Describe please range and median of follow up, number of deceased, and indicate how many patients died from other causes. As follow up was lower than the ideal number of 60 months for many patients, this should be discussed as a limitation of study. Furthermore, the clinical stage has changed over this period. How did authors deal with this?

4. A prognostic factor for overall survival is of little clinical value because most OSCC receives maximum treatment intensity possible in the context of the patient's comorbidities. Parameters that predict local recurrence or nodal or distant metastasis specifically are thus much more likely to be useful clinically to plan treatment. The authors might consider the overall value of this and similar studies in the discussion as so much effort is expended on such studies when clinical oncologists and surgeons usually pay little attention to them at all. In a more philosophical way, can the authors explain why a prognostic factor for overall survival is useful at all?

5. As positive LN detection is sensitive to the experience of the surgeon, were all procedures performed by the same surgeon? If not, how could this affect your results? This should be controlled in your statistical analysis, mentioned in the text, and vigorously discussed.

6. The authors should report how pN was determined. How were the nodes prepared? How many cuts per lymph node? How many sections per lymph node were analyzed? Few cuts/node (likely underestimate) or were they serial sectioned giving a true idea of the presence of metastasis? How about ENE, an important prognostic factor for OSCC (already incorporated into the TNM clinical staging system)? This should be clearly described in material and methods and further discussed. There are 77 pN0 in this cohort, many with advanced tumors.

7. In this cohort, ~90% of the patients were overtreated, because they were N0 at clinical, radiographic and pathological level. However, the long term functional and aesthetic side effects may be experienced by the patients. The authors should add some perspective to this aspect in the discussion, especially what we can glace from this study. Indeed, the authors should add a paragraph in the discussion, stressing the finding’s implications to clinical practice. When should elective neck dissection be performed?

8 Minor comments

- Spell out PS (line 69)

- Define lymph duct invasion (line 86).

- Define neurovascular invasion (line 114).

- Details of the ROC curves should be included in Material and Methods (Data analyses).

Comments on the Quality of English Language

Several minor grammatical and typographical errors are present, and an English language review should be undertaken.

Reviewer 2 Report

Comments and Suggestions for Authors

It’s my pleasure to review the article, whose title is “What are the prognostic factors affecting post-selective neck dissection for patients with cN0 oral cancer?”. Below are my opinions:

 In the first paragraph of the section of Introduction, the authors wrote “Although the observation policy is reasonable for appropriately selected patients [3], lymph node…...” This reference 3 article only included the patients with cancers of tongue and mouth floor, not all sites of oral cancers. If the authors want to emphasize the "observation policy", I suggest they choose other references which covered all sites oral cavity cancers.

In the reference 6, the issue of improved survival by early resection of lymph node metastases was noted mentioned at all and shouldn’t be placed here.

 Only oral maxillary SCC was included in the reference 8, I think the cases in this reference were more like hard palate carcinoma. I still suggest the authors choose other references which covered all sites oral cavity cancers.

 In the section of Introduction (lines 55-58), the authors mentioned that a systematic review and meta-analysis concluded that END did not significantly improve overall survival but significantly reduced the rate of regional nodal recurrence and improved disease-specific survival in patients. It’s cited from the reference 12. In this reference, only cases with early oral tongue cancer were enrolled, not all sites of OSCC. It may not be appropriate.

In line 69, the meaning of “PS1” should be clearly stated. I guess it could be performance status.

 In lines 82-83, I think the authors wanted to express that there were 2 components in this study, one is to study the factors associated with occult lymph node (LN) metastasis of the cN0 neck and the other one is to investigate the prognostic factors related to the treatment outcomes. The description is a little bizarre here. The occult LN metastasis is the purpose of this study to find out which associated factors are related to it; It is not a predictor in itself. The sentence “The primary predictive variables in this study were occult LN metastasis and risk factors for poor prognosis after SOHND” may make the readers confused.

 In line 83, “outcome variables” may not be suitable for both part in this study. For occult metastasis (OM), the factors are not outcome variables, I think they have nothing to do with the outcomes or prognosis and are associated factors related to the occurrence of OM.

 In line 113, the reference for Yamamoto-Kohama classification should be provided.

About the variables for OM, I think some of the prognostic factors are not suitable, such as surgical margin, reconstructive surgery, neck dissection, postoperative radiotherapy, postoperative chemotherapy, local recurrence, neck recurrence, distant metastasis. For example, can you predict if there is OM according to postoperative radiotherapy? This doesn't make sense logically. If the authors agree, please do some revision.

In Table 1 and 3, if the cases in this study are all cN0 neck, the T-classification and stage are the same, and the numbers and the statistical results inside are exactly the same. I suggest that authors just choose one of them.

Why the variables for OM are different from those variables for prognosis? Patients were grouped into four categories based on the BMI value closest to the date of diagnosis according to Centers for Disease Control and Prevention (CDC) guide-lines: underweight (BMI <18.5 kg/m2 ), normal weight (BMI 18.5 – 24.9 kg/m2), overweight (BMI 25 – 29.9 kg/m2), and obese (BMI ≥ 30 kg/m2). BMI is classified in OM items as <18, 18-25, and >25, which is reasonable. Just like we define people over 65 years old as elderly. When analyzing the prognosis, the BMI category was set to 22 as the cut-off point. This is different from the general classification. It is unreasonable to use 22 as the cut-off point just for the purpose of statistical analysis. If the authors found BMI classification (according to underweight, normal weight, and overweight) isn’t significant statistically, it means BMI isn’t an independent prognostic factor related to overall survival.

In the 1st paragraph of Discussion, the authors wrote: In the present study, predicting markers of OM for the SOHND performed in patients with cN0 were NLR (≤1.74 vs. >1.74) and pathological vascular invasion. Moreover, the prognostic factors for SOHND in patients with cN0 OSCC were pN+ level and BMI<22 Kg/m2. These patients require preoperative nutritional therapy, adjuvant chemotherapy, and/or radiotherapy. If a patient’s BMI < 22 kg/m2, he (or she) could be in the status of normal weight or non-malnutrition status. Why do these patients need preoperative nutritional therapy? Besides, what are the criteria for postoperative adjuvant chemotherapy, and/or radiotherapy in this study? Are the criteria related to the Cox regression analyses for overall survival here? The rationale behind this should be clearly explained; otherwise, readers might perceive the author as jumping directly to the conclusion.

In line 225-226, the authors wrote: In the present study, OM was observed in 9 (10.5%) of the 86 patients and previous reports. This sentence doesn’t make sense.

In line 228, a diagnosis is usually made, not detected. I suggested you can describe: the diagnosis of LN metastasis is made…. or the detection of LN metastasis is made…..

In line 230, “previous” here isn’t appropriate. “Preoperative” could be better.

Vascular invasion and neurovascular invasion are both in the Table 1. If neurovascular invasion includes vascular component, why the authors mention the vascular invasion again? In Table 3, nerve invasion was mentioned but neurovascular invasion was absent. Some of the items in Table 1 and Table 3 are inconsistent. Usually perineural invasion (PNI) is a common term in the pathological examination of OSCC. We don’t know if the neural invasion means PNI? I suggest using the same assessment items in both tables as much as possible.

In line 263-264, “……because all patients were cN0, and might nutritional condition was not so bad….” The grammar is wrong. Please correct the wording and grammar of the entire article.

In line 277, “In a meta-analysis, isolated neck……” If the meta-analysis here is the one the authors mention in the last sentence, I think “the” should be used instead of “a”. There are still many errors in it. I won't go into details.

In lines 270-281, the paragraph is about occult neck LM metastasis and the neck recurrence, where the authors describe the findings of their study and compare the differences with previous articles. But it lacks more in-depth analysis and the authors' own insights. In the end of the paragraph, it’s written as “One of the most likely sources of error is underreporting of pathological specimens. Many reported cases of pN0 necks contain OM that cannot be detected by bisection or H&E staining alone [10].” Do the authors have a better idea to solve the problem or improve that or to avoid that? This should be discussed or possible future options proposed.

In line 298, an English letter lowercase b was here (level IIb). Is this level IIb the same as the level IIB here? Why is it a lowercase letter? In line 297-300, the authors wrote “No isolated level IIb metastases were found among patients with positive level IIB, and in the six studies that reported this association, all patients with nodal disease at level IIB had positive level IIA.” About the first sentence, no isolated level IIb metastases were found among patients with positive level IIB; is this finding from this study? But there seemed to be no level IIB nodal metastasis in this study. Or this description is from the six studies? I suggested the authors clarify the description here or the readers will be confused. And please provide the references of the six studies in the sentences.

Overall survival was the evaluation item used in this study. Why didn’t the authors use disease-specific survival (DSS) and/or disease-free survival (DFS) as the evaluation tool? DSS or DFS may be more specifically related to the status of the disease. Authors can refer to these indicators.

Comments on the Quality of English Language

Overall, this research submission is a good idea, but the English level of the writing is average and can be revised.

Reviewer 3 Report

Comments and Suggestions for Authors

This is an interesting study about the predictors of occult metastasis and the prognostic factors associated with the patients with cN0 oral cancer who underwent elective neck dissection.

The paper is well written. However, some issues remain.

How were cut-offs for the variables (age, BMI, lymphocyte count, NLR) selected in table 1?

Why did the authors not evaluated other cut-offs in ROC curves to identify better cut-off values?

Table 2 must include all the variables in the multivariate analysis, not only the significant ones.

All the Kaplan-Meyer curves should be included in the manuscript. Moreover, OS and DFS must analyzed also stratifying for the stage according to AJCC.

Laterality of ND is probably associated to pN. This must be evaluated. If these two parameters were associated, only one should be used for multivariate analysis.

Round 2

Reviewer 1 Report

Comments and Suggestions for Authors

The authors were receptive to my comments and suggestions to improve the manuscript. Thank you. However, I still have some comments after reading the revised version.

1. For some reason, the authors kept in the title and other parts the term clinically N0 (cN0). The fact that all patients were subjected to CT and MRI +/- PET-CT and US is an important (advantage) feature of study, and this should be clearly explored.

2. I do not disagree that elective neck dissection is recommended for early-stage tongue or floor of mouth SCC (by the way, the classical NEJM reference only deals with tongue, floor of mouth and buccal mucosa SCC). The point is that the current cohort has 54% cases of T3 and T4, which are classified as advanced clinical stages. So, it is not clear why the authors are focusing on the early-stage ones. The universal recommendation is that patients with T3/T4 tumors, regardless of N status, should be subjected to neck dissection. The authors should organize the scientific fundamentals of the study (introduction, because they do not explore this feature in the discussion and conclusions) based on their cohort, which is not composed of early-stage tumors only (indeed they represent less than half). If the focus is early-stage, only T1/T2N0M cases should be explored.

3. It is not clear how the authors have discriminated between vascular and lymphatic invasions, but they should correct the text replacing “lymph invasion” by “lymphatic invasion”.

4. Thanks for performing DFS, but the findings should be included in the abstract and conclusions. Furthermore, the fact that N status did not resist the multivariate approach should be discussed, bringing possible explanations for this.

Reviewer 2 Report

Comments and Suggestions for Authors

Regarding the title, I think it doesn’t cover the all aspects of the study and the title appears somewhat ambiguous in meaning. First, there are 2 main parts of the study: occult nodal metastasis (OM) and the prognosis of patients with oral cancers and cN0 neck. The title does not mention the OM at all. Second, I think the prognostic factors won’t affect post-selective neck dissection for patients with cN0 oral cancer. The better expression should be “the prognostic factors associated with the patients with cN0 oral cancer.”

I don’t know what “delated” means through the whole cover letter. I guess it could be “deleted” which was misspelled as “delated”.

According to the authors’ revision L46-47: The article previously referred to [3] by Kelner et al. did not adequately address tumor location. Instead, we have replaced it with the work of Leonardo et al. [4] on "observation policy." The content of the newly added reference 4 by the authors (Leonardo, K.S.K.; Leandro, L.M.; Marco, A.V.K.; Vergilius, J.F.F.; Claudio, R.C. Oral Cancer Treatment: Still an Indication for Elective Neck Dissection? ORL 2018, 80, 96–102) is wrong. It should be Koyama LKS, Matos LL, Kulcsar MAV, de Araújo Filho VJF, Cernea CR. Oral Cancer Treatment: Still an Indication for Elective Neck Dissection? ORL J Otorhinolaryngol Relat Spec. 2018;80(2):96-102.

In line 420, the new reference 16: Matsuzawa, Y.; Tokunaga, K.; Kotani, K.; Keno, Y.; Kobayashi, T.; Tarui, S. Simple Estimation of Ideal Body Weight from Body Mass Index with the Lowest Morbidity. Diabetes Res Clin Pract 1990, 10, 159-164. Actually, this information of this reference is wrong, and the correct content is “Diabetes Res Clin Pract. 1990;10 Suppl 1:S159-164.”

In line 57-60, the authors wrote “A systematic review and meta-analysis concluded that elective neck dissection (END) did not significantly improve overall survival but significantly reduced the rate of regional nodal recurrence and improved disease-specific survival in patients with early stage OSCC.” As the authors revised, a new reference 9 (J Oral Maxillofac Surg. 2019;77(1):184-194) was cited here. In the reference 9, the meta-analysis indicated that regional recurrence might be the main cause of disease-specific death; hence, regional control (management of the neck) must have a positive impact on survival rate. The conclusion of this reference 9 was END substantially decreases recurrences and deaths related to regional recurrences in early-stage SCC of the oral cavity with clinically N0 neck. Judging from the context of the article (reference 9), the survival is more like disease-specific survival, not overall survival. That is to say, it didn’t mention that “END did not significantly improved OVERALL SURVIVAL”. If the authors have evidence supporting the description about overall survival, please add it and cite the reference(s).

In the section of 2.2 Study Variables (line 93-98), the authors only wrote “The primary predictive variables in this study were risk factors for poor prognosis after SOHND.” The “occult LN metastasis” was left out in this section. But some of the variables were also analyzed in the part of analyzing the factors related to occult metastasis (OM) and OM is still one of the main focus in this study. Is it appropriate not to mention anything about OM in this section?

About the factor BMI, I disagree with the authors. If the authors use the traditional classification of BMI (categorical variable) in Table 1, why the BMI is shifted to be a continuous variable in Table 3? Besides, the new cut-off value 22 of BMI is not as consensual as the traditional classification. In the section 3.3 (starting from line 181), the authors mentioned that “The BMI associated with the lowest morbidity in various underlying diseases was calculated to be 22 kg/m2 and ideal body weight is 22 x height (m)2 in Japanese [16,17].” It should not be placed in the section of Results. The section of Discussion could be a better place for it.

In line 291-293, the authors wrote “This study identified no significant difference between BMI <18.5 and ≥18.5 kg/m2 because all patients were cN0, and their nutritional condition was not poor.” NOT POOR? Usually we use BMI as a standard to evaluate a patient's nutritional status. If a BMI less than 18.5 is not considered poor nutritional status, then what is considered poor nutrition?

In line 294-299, the authors wrote “Although the BMI of 22 kg/m2 was within the normal range, a significant difference was observed in the survival curve. …... The BMI ≥22 vs. <22 kg/m2 is a good prognostic marker in the SOHND patients subgroup. Therefore, nutritional therapy before surgery is recommended for improving the prognosis of patients with a BMI <22 kg/m2.” First I’d like to identify that the data about BMI 22 from the reference 17 comes from the reference 16, so it's not necessary to list the both references 16 and 17. In reference 16, the morbidity score was estimated from the ten medical problems, including lung, heart, renal disease, and so forth, not specific to oral squamous cell carcinoma. In a study analyzing the association between BMI and all-cause death in Japanese population, the risk was lowest when BMI was 22.0-24.9 kg/m2 (J Epidemiol. 2019;29(12):457-463). Probably BMI 22 is also a cut-off point for the survival of patients with oral cancer in this study. However, if the authors think that malnutrition means BMI < 22, it’s difficult to persuade the readers. I personally consider patients with BMI < 18.5 are in the status of malnutrition, not BMI < 22.

In line 198-299, the description was “Therefore, nutritional therapy before surgery is recommended for improving the prognosis of patients with a BMI <22 kg/m2.” In my view, BMI of 22 is still WITHIN THE NORMAL WEIGHT RANGE. Why are all patients with BMI less than 22 recommended to receive nutritional therapy? Do the authors mean patients whose BMI is between 18.5 and 22 are suggested to get a nutritional therapy to improve the survival? According to Centers for Disease Control and Prevention (CDC) guidelines, the classification is: underweight (BMI <18.5 kg/m2), normal weight (BMI 18.5 – 24.9 kg/m2), overweight (BMI 25 – 29.9 kg/m2), and obese (BMI ≥ 30 kg/m2). Although this result of cut-off value of BMI 22 is obtained statistically, is it reasonable when interpreted in practice? In a study analyzing the value of BMI 3 months after postoperative adjuvant therapy for patients with locally advance oral squamous cell carcinoma, the cut-off point of BMI was 18.5 and malnourished status was defined as those whose BMI was lower than 18.5 (Laryngoscope. 2012;122(10):2193-2198.). I think it’s more reasonable. If the authors want to use BMI 22 as the cut-off point, the nutrition status must be explained carefully. It is unreasonable to regard a BMI less than 22 as malnutrition.

The meaning of the paragraph about END (line 337-348) is ambiguous. The first sentence “The indication of END for cN0 neck is controversial. Some investigators reported that END could prevent regional recurrence [1, 9]……” Both the two cited references (1,9) refer to early-stage oral cancer patients, not encompassing all stages (Stages 1-4) of oral cancer. This is different from the patients discussed in this article. The references do not necessarily support the findings of the present study. Maybe you can find other references similar to your study or revise the context here. About the aesthetic side effect of END (line 339), is there a reference for that? Please add it if there is one.

The authors set criteria for END and those was described in the section of Materials and Methods (line 70-73). They applied the criteria to treat patients with cN0. In the end of the paragraph (line 346-347), the authors wrote “we suggest considering END for patients who meet our criteria.” Are the criteria (line 347) the same as those in the section of Materials and Methods? If they are, it does not make any sense at all. The indication for END for cN0 in this study ought to be the factors associated with occurrence of OM, not the criteria you used to adopt to treat you patients. Do the authors mean the OM occurs when patients with reconstructive surgery, tumor depth ≥ 4 mm in tongue cancer, and more than segmental resection of the mandible performed when a cervical approach was required for primary resection?

Comments on the Quality of English Language

Does not meet the desired standard.

Reviewer 3 Report

Comments and Suggestions for Authors

Thank you for improving the manuscript.

Round 3

Reviewer 2 Report

Comments and Suggestions for Authors

It’s my pleasure to review the article, whose title is “What are the predictors of occult metastasis and the prognostic factors associated with the patients with cN0 oral cancer who underwent elective neck dissection?” Below are my opinions:

Suggestion 1: Is a question title appropriate?

Regarding the title, the authors have changed the title to “What are the predictors of occult metastasis and the prognostic factors associated with the patients with cN0 oral cancer who underwent elective neck dissection?” First, I think “in the patients ……” could be better than “associated with the patients ……” because with was used twice here. Second, I’d like to raise a question here. Can a research paper title be a question? In academic publications, question sentences are generally not appropriate as titles for several reasons. Titles should be clear and provide information about the research study's focus or scope. Question titles may not effectively convey the key aspects of the research, potentially leading to confusion among readers. Besides, a good title should be descriptive, precise, and accurate, ensuring that it effectively communicates the nature of the research. Question titles may fall short in achieving these criteria. Concerning attracting readers’ attention, an ideal title should also attract attention by being interesting and unique; question titles may not always achieve this goal. In summary, academic titles should prioritize clarity, informativeness, and precision, making question sentences less suitable for this purpose.

Suggestion 2: The expression of the abstract text can be modified.

In line 19-20, the way the abstract is written is a bit strange, it seems that some important factors have been discovered quite conclusively. But judging from the number of patients in this study, there should not be such a strong conclusion. It can be expressed more conservatively and the word expression can be revised.

Suggestion 3: Occult nodal metastasis, early oral squamous cell carcinoma vs. full stages of oral squamous cell carcinoma. Is the number of the patients enough?

The indication for elective neck dissection in cN0 oral cancer patients has long been a subject of debate. Mainly because among patients with cN0 neck, some patients have occult nodal metastasis (OM) but some don’t. I think it’s meaningful to investigate the factor(s) associated with OM in cN0 oral cancer patients. When it comes to this, another interesting subject arises: whether there is OM is a hot topic among many researchers, especially in patients with early-stage oral cancer. In the first sentence of the first and second paragraphs in the section of Introduction, the authors also mentioned “early” oral squamous cell carcinoma. I also found that among all 23 cited references, 4 of them are related to the topic of early-stage oral cancer. If the authors want to change the study and put emphasis on the patient with “early oral cancer”, then keep this direction. If the authors would like to evaluate the patients with all stages of oral cancer, just let go and do it. However, in terms of the number of patients in this study, if only patients with early-stage oral cancer are discussed, the number may be quite insufficient, making the conclusion weak and unconvincing.

Suggestion 4: About the predictive variables, is “primary” or “secondary” appropriate here?

Regarding the predictive variables (line 97-103), primary predictive variables are central factors studied in research to answer a specific question. In clinical research, the primary outcome is the most relevant variable addressing the research question, ideally patient-centered. Secondary predictive variables are additional factors studied alongside primary variables. In the context of attendance, for example, secondary predictive variables are factors associated with attendance but not the main focus. They are considered in relation to the primary variable, and their impact is analyzed through adjusted odds ratios and confidence intervals. But in this study, excluding those not suitable, the factors in Table 1 are the same as those in Table 3, they aren’t primary or secondary; they’re the same. I don’t think it’s appropriate to use primary or secondary to describe the predictive factors here.

Suggestion 5: Are the factors associated with OM aslo prognostic factors? If not, don’t use prognostic markers.

In line 182-183, the authors wrote: “These results indicated that NLR (≤1.74 vs. >1.74) and vascular invasion are good prognostic markers of OM. Logistic regression analysis of the factors related to OM was done and the results were shown on Table 2. I think the end point of OM is occurrence, not prognosis of treatment for oral cancer. It may not be appropriate to use “prognostic markers” here; “independent associated factors” could be better.

Suggestion 6: The cited references (7, 8) don’t seem to be related to the context.

In line 343-344, the references was corrected to [7, 8] by the authors. The authors wrote “The indication of END for cN0 neck is controversial. Some investigators reported that END could prevent regional recurrence [7, 8]…” In reference 7 (Cancers (Basel). 2022;14(17).), the aim of the study was to compare survival curves of patients suffering from lymph nodal metastases in a preoperatively N+ neck with those suffering from OM. The conclusion of this article is the presence of occult metastasis could lead to decreased survival and could be a burdening factor requiring treatment escalation and a more aggressive follow-up. The indication of neck dissection is early oral cancers is also argued in this article, but the role of neck dissection to prevent regional recurrence didn’t seem to be included. In the reference 8 (Oral surgery, oral medicine, oral pathology and oral radiology. 2017;124(1):32-36.), the object of this study is analysis of the incidence and clinical relevance of neck failure after elective neck dissection (END) in patients with oral squamous cell carcinoma. In the 188 patients with OSCC s/p neck dissection, the disease-specific survival was 95.4% in pN(─) patients and 89.1% in pN(+) patients. There was no statistically significant difference between the survival rate in pN(─) and pN(+) patients. Whether performing neck dissection can prevent regional recurrence was not mentioned at all. Having read these 2 articles, it seems that in these two papers, there is no mention of surgery preventing local recurrence. Please choose the corresponding reference(s) to support the concept you described.

Suggestion 7: Is pathological vascular invasion from biopsy specimens or whole tumors? Which one is better?

In line 350-353, the authors wrote “We found that OM occurred in 10.5% of cases, and NLR (≤1.74 vs. >1.74) and vascular invasion are good markers to detect OM after SOHND in patients with cN0. Therefore, we suggest considering END for cN0 patients who meet our results; NLR (≤1.74 vs. >1.74) and vascular invasion in biopsy specimen.” First, why is it in the biopsy specimen? I thought the pathological findings in Table 1 were from the excised tumors. Am I wrong? A biopsy specimen usually can’t represent the status of a whole tumor. If pathological vascular invasion was not found in the specimen of a biopsy due to the sampling error, it could be not robust to use the findings in a biopsy sample in the analysis of occurrence of OM. If all the patients received total excision of the oral tumors in your study, why don’t the authors use the findings of the excised tumors? The findings of whole tumors are far better than the biopsy specimens, aren’t they?

Comments on the Quality of English Language

fine
